# An ecological network approach to predict ecosystem service vulnerability to species losses

Aislyn A. Keyes [1✉], John P. McLaughlin[2], Allison K. Barner[3] & Laura E. Dee[1]

Human-driven threats are changing biodiversity, impacting ecosystem services. The loss of one species can trigger secondary extinctions of additional species, because species interact–yet the consequences of these secondary extinctions for services remain under-explored. Herein, we compare robustness of food webs and the ecosystem services (hereafter 'services') they provide; and investigate factors determining service responses to secondary extinctions. Simulating twelve extinction scenarios for estuarine food webs with seven services, we find that food web and service robustness are highly correlated, but that robustness varies across services depending on their trophic level and redundancy. Further, we find that species providing services do not play a critical role in stabilizing food webs – whereas species playing supporting roles in services through interactions are critical to the robustness of both food webs and services. Together, our results reveal indirect risks to services through secondary species losses and predictable differences in vulnerability across services.

[1] Department of Ecology and Evolutionary Biology, University of Colorado, Boulder, CO, USA. [2] Depeartment of Ecology, Evolution, and Marine Biology, University of California, Santa Barbara, CA, USA. [3] Department of Biology, Colby College, Waterville, ME, USA. ✉email: Aislyn.Keyes@colorado.edu

Anthropogenic threats, such as climate change and species' overexploitation, are degrading ecosystems and their capacity to provide ecosystem services[1,2]—nature's contributions to human well-being. Anticipating how these threats that cause species losses will impact ecosystem services poses an urgent science and policy challenge[3–5]. Our ability to predict how threats from species losses will affect ecosystem services is complicated by the fact that species interact in complex networks to perform the functions underpinning ecosystem services[6–10].

Despite the complex ecological interactions involved in producing ecosystem services, assessments that quantify ecosystem services generally focus on the species directly providing services (e.g., bees that pollinate crops) and direct threats to those species (e.g., disease affecting bees). However, species that directly provide ecosystem services—hereafter, "ecosystem service providers" following Kremen (2005)—interact with other species. These interactions can support (e.g., acting as resources) or inhibit (e.g., acting as predators) their ability to provide services[11]. For example, the giant panda (*Ailuropoda melanoleuca*) relies solely on bamboo (e.g., *Bambusa sinospinosa*) as its food source, which supports its ability to provide cultural ecosystem services. Species that support ecosystem service providers—referred to as supporting species hereafter—are also impacted by threats, including land conversion decreasing bamboo availability. Such impacts can trigger species losses, which can lead to the loss of additional species that depend on the initial species lost[12–14]. These additional species losses are known as secondary or "cascading" extinctions[15]. Secondary extinctions pose an indirect and underexplored threat to ecosystem services[16]. By focusing only on direct threats to ecosystem service providers, ecosystem service assessments may miss the indirect threats that occur through species losses. This knowledge gap constrains our ability to anticipate changes in ecosystem services from threats that cause species losses in food webs and consequent species losses[8].

Insights from network ecology can advance our knowledge about direct and indirect threats to ecosystem services posed by species losses. In particular, robustness studies quantify indirect effects of secondary extinctions in food webs, wherein food web robustness measures food web response to primary and secondary species losses[7,12,17]. Robustness studies have advanced understanding of the factors that determine food web responses to species losses, finding that food web robustness largely depends on (1) network structure[12,15,18,19] and (2) the order that species are removed in species loss scenarios[12,20]. This order of species loss is determined by the type of threat impacting species in a food web[21,22]. Further, if important species playing stabilizing roles in the food web are directly threatened and removed first, the food web may be less robust. For example, the removal of highly connected species causes many secondary extinctions and rapid food web collapse in grassland ecosystems[12]. Findings like these could have important implications for the extent that species losses could also trigger losses of ecosystem services indirectly—which we call ecosystem service robustness. However, to our knowledge, this methodology and the insights about food web robustness have not yet been applied to understand direct and indirect threats to ecosystem services[17].

An outstanding question is whether ecosystem services are more or less robust to species losses than the food webs upon which they depend, and how this answer varies by the ecosystem service studied. The direct and indirect risk from species losses to a particular ecosystem service could be similar, lower, or higher than to the food web. The extent to which this risk from species losses is similar for the food web and ecosystem services depends on whether the species lost are critical to both the food web and the ecosystem service. If species lost from ecosystems either provide services directly or support those species that do, species losses could also lead to losses in ecosystem services[23]. Most species play some role in services (Fig. 1d)—and most often in supporting roles—which suggests that threats to food webs and consequent secondary extinctions could also degrade ecosystem services (Fig. 2b)[10,11]. For instance, a threat that causes the loss of habitat-forming plants could cause the secondary loss of fish targeted by a fishery. Alternatively, impacts to food webs that trigger species losses and secondary extinctions may not ultimately impact ecosystem services—or may differ across services—if the lost species are not ecosystem service providers or their critical supporting species (see Figs. 1 and 2c, d). Finally, the risk of ecosystem service loss could be higher than the risk of food web collapse—when threats cause losses of ecosystem service providers that most other species do not depend on (Fig. 2a). Which of these scenarios is most likely to occur remains unknown, and likely depends on when species are lost and their role in the focal ecosystem service. We investigate the relationship between risk to food webs and ecosystem services from species losses here through an extension of robustness analyses from network ecology.

Here, we aim to understand the extent that species losses in food webs can pose indirect threats to ecosystem services, asking three questions: (1) Is food web robustness correlated with or decoupled from ecosystem service robustness across different sequences of species extinctions? (2) Does the robustness to species losses vary across ecosystem services? (3) Are the species that contribute to ecosystem services, either directly or in supporting roles, critical to food web persistence (i.e., robustness)? We address these questions by simulating 12 extinction sequences on three empirical, estuarine food webs with ecosystem services added (Fig. 1a–c). We compare commonly used extinction sequences from food web studies (e.g., most to least connected species[15,16,20,24–26]), to novel sequences for both food web and ecosystem service robustness. We first hypothesize that the robustness of a food web and its ecosystem services are positively related, but that this relationship depends on the order of species lost, and whether species removed play an important stabilizing role (see Fig. 2 for predictions). To that end, we hypothesize that individual ecosystem services will have varying responses to species losses[17] and that those provided by many species (i.e., higher redundancy[15]) or with lower trophic levels (as in refs. [9,17]) will be more robust. Here, we have sampled ecosystem services that vary in both their trophic level and redundancy to investigate this question using robustness analysis. Further, we hypothesize that ecosystem service providers critical to ecosystem services are not critical for food web robustness compared to ecosystem services, but that supporting species are important to both food web and ecosystem service robustness. We find that food web and ecosystem service robustness are highly correlated, but that robustness varies across ecosystem services depending on their trophic level and redundancy. Further, we find that ecosystem service providers do not play a critical role in stabilizing food webs—whereas species playing supporting roles in ecosystem services are critical to the robustness of both food webs and ecosystem services. Through integration of food web theory and ecosystem service science, this work contributes to our understanding of ecosystem service vulnerability to both direct and indirect threats.

## Results

**Food web robustness is strongly and positively correlated with ecosystem service robustness.** Food web robustness was positively correlated with ecosystem service robustness ($r_s[36] = 0.884$, $P = 9.504e–13$), supporting our hypothesis that the two values are related (Fig. 3). Furthermore, we found support for many of our predictions about the relationship between the food

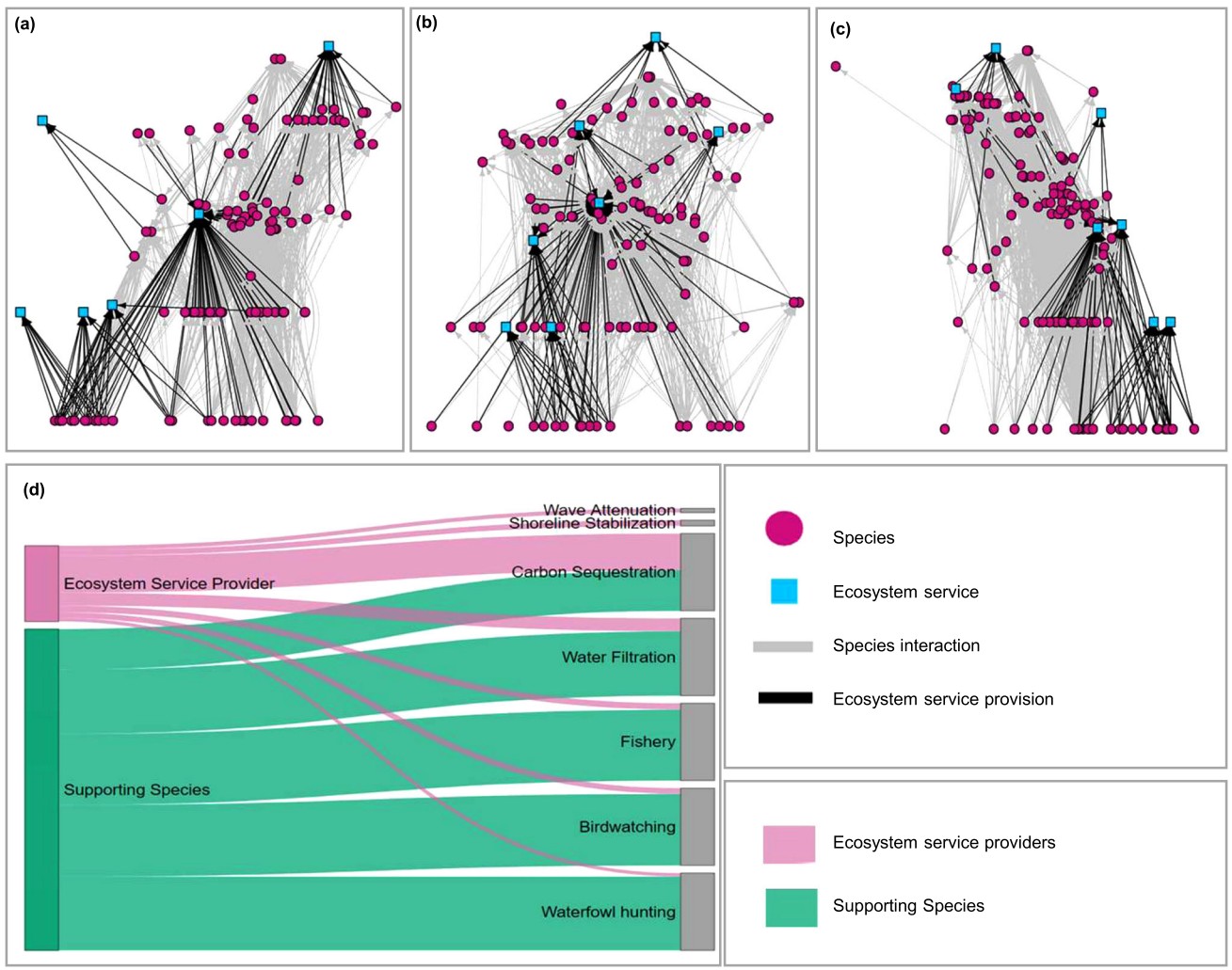

**Fig. 1 All species play important direct and supporting roles in ecosystem service provision.** Panel 1 (top): network visualizations of the (Hechinger et al.[52]) data after initial filtering and adding the seven ecosystem services: water filtration, shoreline stabilization, carbon sequestration, wave attenuation, waterfowl hunting, bird watching, and fishing. Each of the three networks (top) represents one of the three salt marsh systems, organized vertically by trophic level: **a** Bahia Falsa de San Quintin, Baja, Mexico (122 species, 6 ecosystem services (no fishery), 1060 species—species links, and 137 species—service links), **b** Carpinteria Salt Marsh, California, USA (107 species, 7 ecosystem services, 1015 species—species links, and 105 species—service links), and **c** Estero de Punta Banda, Baja, Mexico (136 species, 7 ecosystem services, 1680 species—species links, and 101 species—service links). Panel 2 **d**: Species can provide ecosystem services directly (ecosystem service providers, pink), they can support ecosystem service providers (supporting species, green), or they can play no role in service provision (not ESP, yellow). Most species play some supporting role in service provision. **d** shows the number of species that are ecosystem service providers and supporting species, for Carpinteria Salt Marsh.

web and ecosystem service robustness (Figs. 2 and 3). The results were not sensitive to different $x$-axis calculations (Supplementary Methods and Discussion).

The strong, positive correlation between the food web and ecosystem service robustness was consistent across two of the three types of sequences. The correlation was the strongest for topological sequences ($r_s[12] = 0.944$, $P = 2.2e-16$), followed by the ecosystem service sequences ($r_s[18] = 0.825$, $P = 2.01e-05$). The threat-based sequences yielded a strong, positive, but insignificant correlation ($r_s[6] = 0.759$, $P = 0.080$), likely due to the sample size.

**Individual ecosystem service robustness varies with trophic level and redundancy.** Individual ecosystem service robustness ($R_{indiv}$) was associated with both trophic level and redundancy across all models ($P < 0.001$, Supplementary Table 8-A). When all sequences are included, $R_{indiv}$ increases by 0.3% (SE ± 0.1%) with redundancy and decreases by 6% (SE ± 2.5%) with trophic level

(Supplementary Table 8-A). However, when low trophic level species are removed first (i.e., most-to-least connected and ecosystem service providers high-to-low biomass), $R_{indiv}$ increases by 1.4% (SE ± 0.8%, Supplementary Table 8-B) and 17.1% (SE ± 3.2%, Supplementary Table 8-C), respectively, with trophic level.

**Ecosystem service providers are not critical for food web robustness.** The removal of ecosystem service providers (high-to-low and low-to-high biomass) in each of the three systems, caused the secondary loss of all ecosystem services. However, the removal of ecosystem service providers did not cause a complete collapse in the food web (i.e., there are species remaining in the food web following the removal of all target species, Figs. 4b and 5Ia, b).

**Important supporting species are critical for both food web and ecosystem service robustness.** Removing supporting species —in order of their importance to ecosystem services—had the largest impact on the food web and ecosystem service robustness of

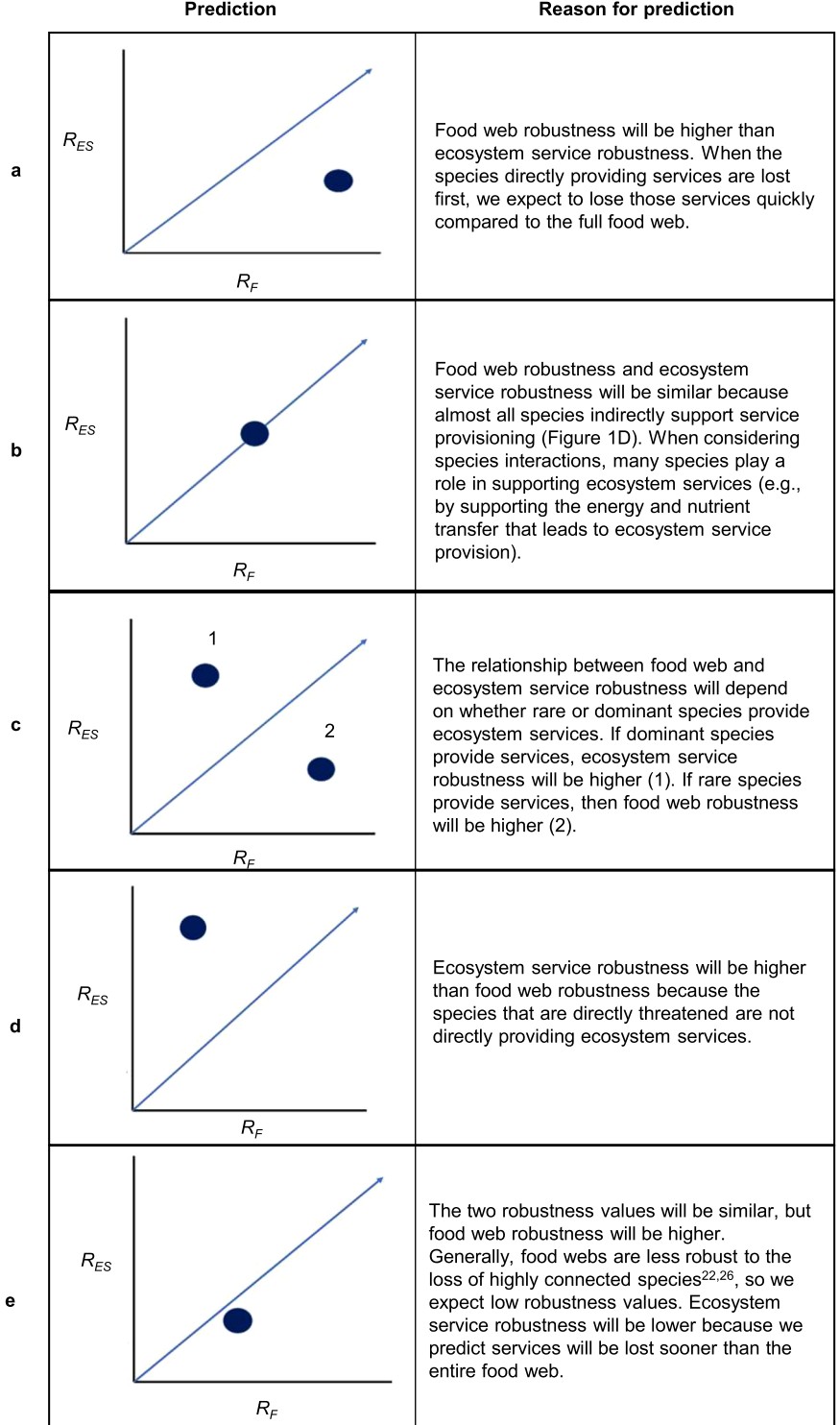

**Fig. 2 Predictions for the relationship between food web robustness ($R_F$; the tolerance of a food web to species loss) and ecosystem service robustness ($R_{ES}$, the tolerance of ecosystem service nodes in a network to species loss).** In each figure ((**a**–**f**), middle column), the x axis is food web robustness, the y-axis is ecosystem service robustness, and the blue line represents a 1:1 relationship between the two ($x = y$). The dots show the relationship we expect between the two robustness values under each sequence of species loss, shown in different rows: **a**–**e**. Thus, if the robustness of the ecosystem services is highly correlated with the robustness of the food webs, the blue dot would fall on the line.

any sequence (Figs. 3 and 5I). Both the food web and ecosystem services collapsed earlier when important supporting species were removed than for any other extinction sequence. However, the removal of supporting species from least-to-most important did not result in a similar collapse of the food web or ecosystem services.

**Weighting species' contributions to ecosystem services decreased their robustness**. Both aggregate (Supplementary Fig. 8) and individual (Fig. 6a) ecosystem service robustness values decreased when considering disproportionate contributions of species to ecosystem services. Yet, the unweighted and

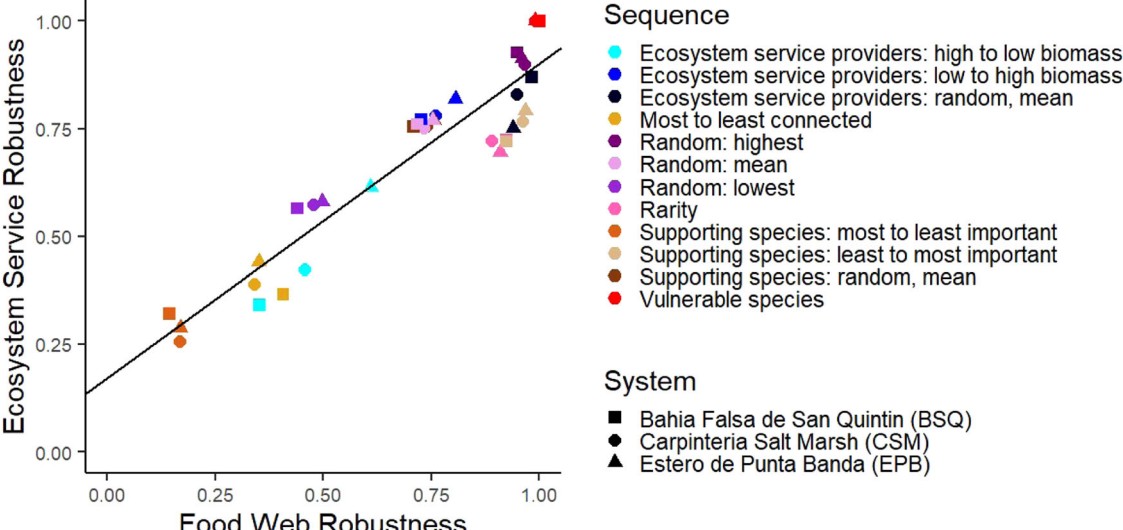

**Fig. 3 Food web and ecosystem service robustness are highly and positively correlated ($r_s$[36] = 0.884, $P$ = 9.504e–13).** We ran a two-tailed Spearman's correlation test on the food web and ecosystem service robustness for all 3 systems and all 12 sequences. The black line is a regression line. The shape of data points represents the salt marsh system: Bahia Falsa de Sant Quintin (square, Fig. 1c), Carpinteria Salt Marsh (circle, Fig. 1a), and Estero de Punta Banda (triangle, Fig. 1b). The color denotes the sequence of species removals: ecosystem service providers (removed by biomass and randomly), supporting species (removed by high-to-low importance, low-to-high importance, and randomly), most-to-least connected, randomly (reporting: highest, lowest and mean, $n$ = 1000 randomizations), rarity (removed by relative abundance—from least to most abundant), and vulnerable species (removed from least to most abundant).

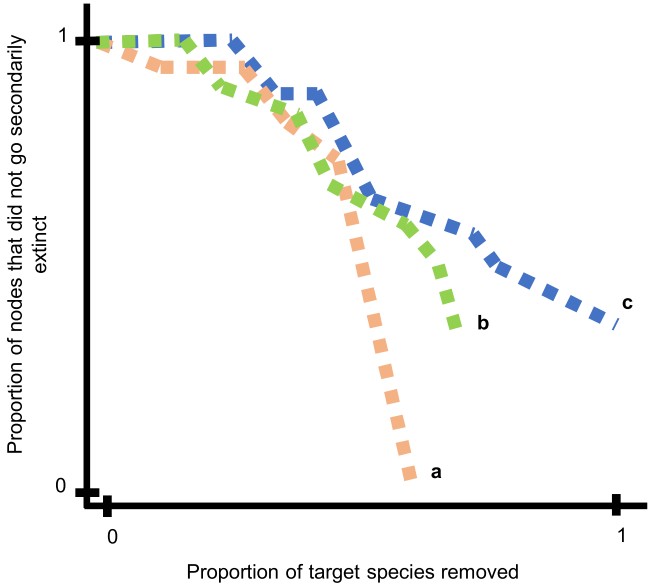

**Fig. 4 Illustrating robustness calculations.** As species are removed (i.e., primary extinctions) secondary extinctions occur when species no longer have resources (in-degree = 0). Primary extinctions are tracked on the x axis and secondary extinctions on the y axis as proportions and robustness is the area under the curve. If all target species are removed, the x axis will reach x = 1 (e.g., line c). If a subset of the target species is secondarily lost (e.g., lines a and b), the line will not reach x = 1. If the line reaches y = 0, all secondary losses are realized (e.g., line a). If the line does not reach x = 1 or y = 0, that implies that (1) some target species went secondarily extinct, and/or (2) there are still species remaining in the food web (e.g., line b).

weighted ecosystem service values were strongly correlated for both the aggregate ($r_s$[36] = 0.760, $P$ = 7.439e–08) and individual ($r_s$[36] = 0.918, $P$ = 2.2e–16) ecosystem service robustness values. Further, the strong, positive relationship between the food web

and ecosystem service robustness remained ($r_s$[36] = 0.885, $P$ = 7.395e–13, see Supplementary Fig. 9).

## Discussion

Understanding direct and indirect threats to ecosystem services from species losses is a key question in the ecology of ecosystem services[17,27]. Robustness is one way to measure how species losses and associated secondary extinctions will impact food webs[12,28,29]; and here, we investigate the consequences of these secondary extinctions for ecosystem services. Extending robustness analyses to ecosystem services, we further our understanding of how indirect threats from secondary species extinctions will impact ecosystem services. Here, ecosystem service and food web robustness values were strongly and positively correlated across scenarios of species losses ($r_s$[36] = 0.884, $P$ = 9.504e–13), suggesting that food web robustness can predict ecosystem service robustness, at least in these salt marsh systems (Fig. 3). The methods used in this study are applicable to a variety of trophic interactions (e.g., predation, micropredation, and parasitism) and ecological networks.

We found that ecosystem service providers—integral to ecosystem service robustness—do not have an equivalent role in food web robustness, yet that important supporting species were critical to the robustness of both. This finding reveals two things. First, ecosystem service vulnerability may be greater than anticipated because many species indirectly support their provision (Figs. 1d and 7, sequence I). Here, removing important supporting species caused the rapid collapse of ecosystem services (lowest robustness values for all systems; Fig. 7, sequence I). However, removing supporting species from least-to-most important and randomly did not cause a rapid collapse in either the food web or ecosystem services in aggregate (Fig. 7, sequences J, K). This implies that while many species indirectly support ecosystem services, the sequence with which they are lost determines the magnitude of indirect risk. Second, this finding reveals the disconnect between ecosystem service providers' role in providing services and stabilizing food webs. In particular, removing ecosystem service providers (Figs. 2a and 7, sequences B–D) caused the collapse of ecosystem services, but not the food web (i.e., species remained in

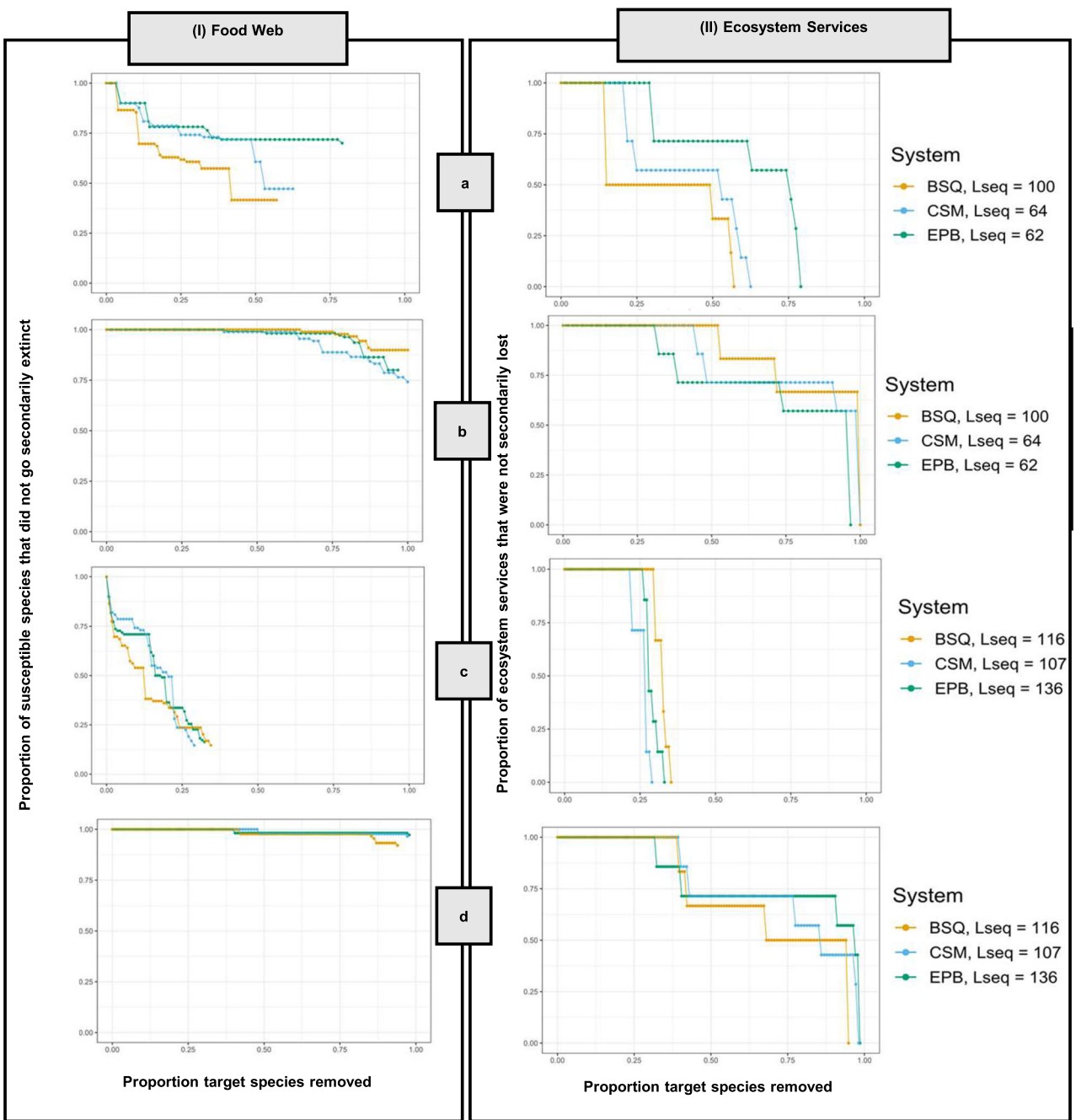

**Fig. 5 Food web (I) and ecosystem service (II) responses to primary species removal result in different rates of secondary losses.** Each box shows the sequential loss of species and/or ecosystem services for 4 of the 12 species loss sequences (see Supplementary Fig. 7 for the remaining sequences): **a** Ecosystem service providers, high to low biomass, **b** ecosystem service providers, low to high biomass, **c** Supporting species, most to least important, and **d** Supporting species, least to most important. The three colors show the three salt marsh systems: Bahia Falsa de San Quintin (BSQ, gold), Carpinteria Salt Marsh (CSM, blue), and Estero de Punta Banda (EPB, green). $L_{seq}$ indicates the length of each sequence (the denominator in the x axis). Note that not all sequences are the same length.

the food web, Fig. 4). In contrast, the removal of important supporting species resulted in many secondary extinctions and the food web collapse. Overall, our results reveal the important role of supporting species in providing and maintaining robust ecosystem services and food webs.

The order of species losses (extinction sequences) affects the robustness of both food webs and ecosystem services (Figs. 3 and 4), consistent with past work on food webs[18,20,30] and plant-pollinator networks[31]. We found that both food web and ecosystem service robustness were lower for the removal of most-to-least connected

species than for the random removal of species (Figs. 3 and 4), consistent with the previous studies[12,15,20]. When we compare commonly studied species loss sequences (e.g., removing species randomly, and ordered based on the most to least connected) with realistic sequences (i.e., removing species ordered by rarity), we see that both food web and ecosystem service robustness values are similar for species lost based on rarity and random (Fig. 7F, H). This suggests that commonly studied sequences (e.g., random removals) may actually parallel what we expect to see under realistic, threat-based species losses.

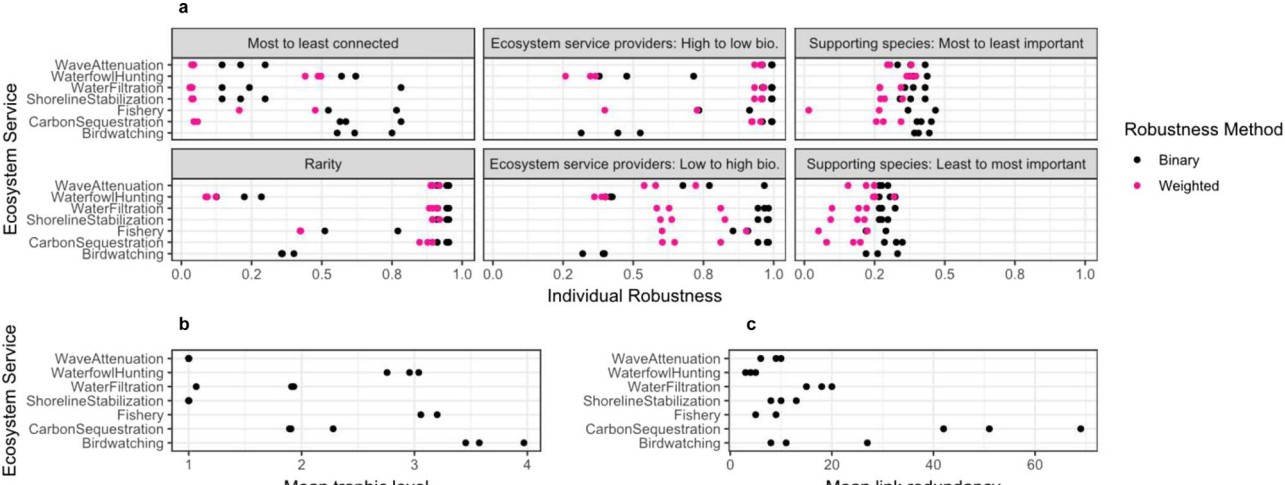

**Fig. 6 Individual ecosystem service robustness varies by trophic level and link redundancy. a** Individual ecosystem service robustness across the three salt marsh systems and sequences: most to least connected, ecosystem service providers: high-to-low biomass, supporting species: most-to-least important, rarity, ecosystem service providers: low-to-high biomass, supporting species: least-to-most important. Individual ecosystem services respond differently to species across sequences. Pink data points represent individual ecosystem robustness values when calculated based on presence–absence of a species and black data points represent individual ecosystem service robustness values when calculated based on biomass. Birdwatching was excluded from the weighted method extension because the relationship between species and service is not proportional to biomass. **b** Trophic level of each ecosystem service across systems. **c** Link redundancy of each ecosystem service across systems.

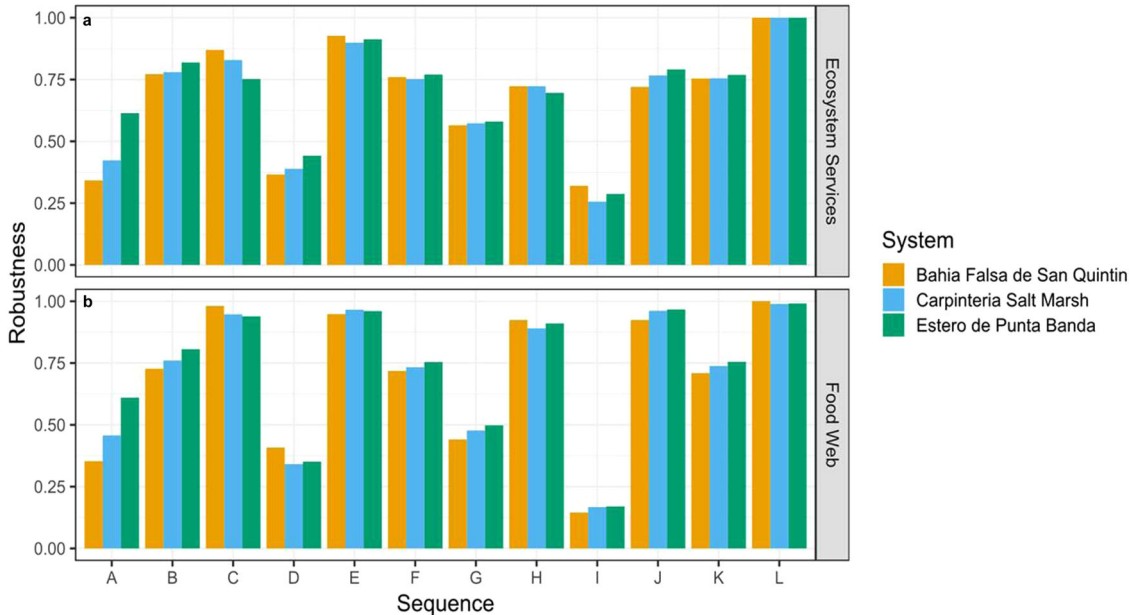

**Fig. 7 Ecosystem service and food web robustness vary across species loss sequences. a** ecosystem service robustness values and **b** food web robustness across the following species loss sequences: (A) Ecosystem service providers: high-to-low biomass, (B) ecosystem service providers: random mean, (C) ecosystem service providers: low-to-high biomass, (D) most-to-least connected, (E) random; maximum, (F) random; mean, (G) random; minimum, (H) rarity, (I) supporting species: most to least important, (J) supporting species: least-to-most important, (K) supporting species: random mean, (L) vulnerable species.

Beyond certain thresholds of species losses, we observed that ecosystem services rapidly collapse (Fig. 5IIa, c), Supplementary Fig. 7 (II)a. The methods used here could be used to shed light on when and how thresholds in food webs that result from species losses may impact certain ecosystem services—a key identified research priority in ecosystem service research[32]. For example, in the important supporting species sequence (Fig. 5IIc), ecosystem services are not lost until ~20% of the species are removed, at which point the ecosystem services collapse rapidly. When tracking which ecosystem services are lost in the sequence of species losses, we observe that some ecosystem

services are more robust to species losses than others. Specifically, ecosystem services that are provided by more species (i.e., higher redundancy) generally have higher robustness than those that are provided by few species (Fig. 6), consistent with studies examining relationships between biodiversity and ecosystem functioning within a single trophic level[33]. Further, ecosystem services at higher trophic levels tend to be less robust than those at lower trophic levels (Fig. 6) —consistent with prior ecosystem service studies in marine and grassland ecosystems that consider species interactions[9,17]. While we find results consistent with prior research that trophic level and

redundancy are associated with differences in ecosystem service responses to species losses, additional research is needed to assess other potential factors influencing variation in the robustness of multiple services to species losses.

Our approach, along with prior robustness studies, makes several assumptions that could over- or underpredict risk from species losses and secondary extinctions. Topological—or structural—robustness analyses are a bottom-up estimate of network stability and do not account for top-down effects, competitive dynamics, or the potential for adaptive dynamics like prey-switching following species losses, i.e., "rewiring"[26,31]. Not including these dynamics can both underestimate[20,34,35] or overestimate secondary species losses[31,35]. On one hand, a lack of top-down dynamics underestimates secondary extinctions and could overestimate robustness[20,34]. For example, the loss of a top predator in systems with strong top-down effects can have indirect consequences for food webs and services through trophic cascades[36,37], which were not accounted for in our models that omit dynamics. The omission of top-down control could also have implications for the importance of ecosystem service providers in stabilizing food webs. Some salt marshes—with the exception of those in our study—are strongly controlled by top-down regulation[38], which could imply that higher trophic level ecosystem service providers are more important in stabilizing food webs than we find here. On the other hand, local species losses may not trigger secondary losses if consumers are able to adapt and change prey species, or if new prey species are introduced[26,31]. Models that incorporate this rewiring find fewer secondary extinctions (i.e., higher robustness) in food webs[26], yet the consequences of rewiring have yet to be addressed for ecosystem services and their robustness, to our knowledge. Further, like previous robustness studies (see refs., [24,39]), our model does not account for the recolonization of previously lost species, which could occur if the system is well connected to other habitats. The impacts of competition on food web dynamics can be important[26,40] but are similarly not accounted for in topological approaches, but arise when two species are indirectly linked—e.g., by competing for the same resource or sharing a consumer—and one competitor is lost[26,40]. Finally, many species undergo ontogenetic diet shifts, and as such, operate as serial specialists rather than generalists. Our analysis did not take this stage structure into account, but previous studies have demonstrated its importance to robustness estimates, especially when parasites are included[41]. Future research addressing how ecosystem service robustness depends on model choice is needed to address these limitations, and accounting for these dynamics in future research could improve our understanding of the relative importance of these processes for ecosystem services.

While the ecological processes outlined above could determine food web and ecosystem service responses, we used a topological approach for several reasons. First, dynamic food web models require specifying many parameters, interaction strengths between species, and functional forms: all of which govern dynamics but are highly uncertain[14,42] and costly to resolve in most real systems. Using a topological approach does not require as much systems knowledge, only information about food web structure; as such, these analyses have been applied widely to a variety of ecosystems[17,43], including for analyzing food web robustness[12,30]. In contrast, most prior robustness studies that incorporate dynamics use simulated food web data based on generic network models[20,34,44]. Here, we opt to use a topological approach in order to demonstrate, in real systems, the indirect threats to ecosystem services that can arise when considering their supporting species and interactions, and the methods that can be applied to understand these threats. Second, incorporating ecosystem service dynamics in food webs, and their functional relationships with species, is an important frontier in ecosystem service science. For most services and systems, species-specific or functional group-specific contributions to ecosystem services and their dynamics are not well known, except for services that depend directly

on population harvest (e.g., for services like fisheries that are measured and valued based on biomass of particular species, like bluefin tuna). But, for services like shoreline protection, species-specific, direct contributions and the dynamics of those relationships through time are not well resolved, and can be highly complex (e.g., dependent on diversity, abundance, and identities of species)[45–47]. To begin addressing the nuances involved in ecosystem service provision in a food web context, we compared the ecosystem service robustness results when species' contributions are equal versus weighted according to their biomass[48,49]. We found that ecosystem services were less robust to species losses when considering species' unequal contributions to services than they were when assuming all species contribute equally (Fig. 7a and Supplementary Fig. 8). This is because each time an ecosystem service provider is lost, the proportion of ecosystem services remaining decreases. In the unweighted calculation, the proportion of ecosystem services remaining only decreases when it loses all of its ecosystem service providers. While our study does not incorporate the important dynamics outlined above, it makes a first step toward understanding the impacts of secondary extinctions for ecosystem services.

Combining food web ecology with ecosystem service science provides a pathway to understanding the underlying role of species interactions in mitigating direct and indirect threats to ecosystem services, as shown here. We found a strong, positive relationship between food web and ecosystem service robustness, suggesting that food web robustness and insight from network approaches can help predict ecosystem service responses to species losses. In addition, we found that individual ecosystem services with higher redundancy and lower trophic levels are more robust to species losses. Our results highlight the contribution that supporting species make to sustaining ecosystem services. Further, by considering species interactions, we find that secondary extinctions pose indirect threats to services that lead to increased vulnerability. Considering the limited resources available for conservation and the tendency of studies to focus on individual threatened species instead of key interactions[10,50,51], species interactions represent a critical gap in our knowledge of ecosystem services. Omitting these interactions, and the potential for indirect threats to ecosystem services can lead to unintended outcomes (see ref. [10]) further deteriorating ecosystems and the benefits they provide to society. Our incorporation of ecosystem services in ecological networks to measure robustness is a step toward understanding how species interactions and secondary species losses can lead to indirect risks to ecosystem services.

## Methods

**Dataset**. We integrate ecosystem services into food webs from three coastal salt marsh ecosystems[52]: Carpinteria Salt Marsh in Santa Barbara, CA, USA, and Estero de Punta Banda and Bahia Falsa de San Quintin in Baja, Mexico (Supplementary Fig. 1). These highly resolved networks contain detailed information on species nodes (e.g., consumer strategy, body size, and abundance) and trophic links[52]. Prior to including ecosystem services in the food webs, we filtered the species to include only adult stages while excluding nonliving resources (i.e., detritus) and parasites (Fig. 1; Supplementary Data 1, see Supplementary Discussion for the results of including parasites).

**Assigning ecosystem services to ecological networks**. We incorporated seven ecosystem services into these food webs that spanned multiple trophic levels and were provided by varying numbers of species (i.e., redundancy[25]): birdwatching, carbon sequestration, fishery, shoreline stabilization, water filtration, waterfowl hunting, and wave attenuation (Fig. 1). To do so, we identified species that directly provide services, which we call ecosystem service providers, by reviewing primary literature and government reports (e.g., California Department of Fish and Wildlife[53]). We used the methods described in ref. [49] to assign service provision to each species for carbon sequestration, water filtration, wave attenuation, and shoreline stabilization (see Supplementary Methods). Expanding on[49], we consider ES provision at the species level (see Supplementary Methods) and also consider species contributions to birdwatching, waterfowl hunting, and fishing, as described below. Species–service links were recorded as present, ("1" for ecosystem service providers) or absent ("0" for all other species), forming a network containing nodes that are species and nodes that are ecosystem services (Fig. 1, following refs. [10,49]). Thus, this network contains two types of links.

Species–species links describe consumptive links (i.e., one species eats the other species). Species–service links describe ecosystem service provisioning (i.e., the species directly provides that service) for each of the seven ecosystem services. Supplementary Table 1 presents the number of species providing each ecosystem service.

**Birdwatching**. We assigned the birdwatching service to particular species using eBird, an online citizen science program[54]. We assigned the birdwatching service to species based on their rarity—which is valued disproportionately in wildlife viewing[55]. We established a rarity threshold based on count data (see Supplementary Methods) from the eBird Basic Dataset[54] using the R package auk v. 0.4.1[56]. We calculated relative frequency for each species (i.e., the percent of total counts for all species in each estuary, see Supplementary Methods). We assigned the birdwatching ecosystem service provision (i.e., link between bird species and ecosystem service = 1) to bird species with a relative frequency below 0.5%.

**Waterfowl hunting**. Waterfowl hunting services are based on information provided by the California Fish and Game Commission and the Mexican Secretariat of Environment and Natural Resources (SEMARNAT[57]; see Supplementary Methods). While waterfowl may not be hunted in these specific salt marshes, they are likely to move across the landscape to areas that are hunted. If evidence existed that a particular species is hunted, we assigned the species a link to waterfowl hunting.

**Fisheries**. We identify species that contribute to fisheries using fisheries reports (e.g., Diario Oficial de la Federación 2017) and regulatory documents (e.g., California Department of Fish and Wildlife[53]; see Supplementary Methods). While not all fisheries take place within the estuaries, fish are likely to move in and out of these habitats over their lifespan. If evidence existed that a fishery exists for a species (e.g., regulations for a species-like season length), we assigned the species a link to fisheries.

**Calculating individual ecosystem service trophic level and redundancy**. To explore whether and why robustness varies across the seven ecosystem services, we calculated the trophic level and redundancy for each ecosystem service. We calculated the trophic level of ecosystem services as the mean trophic level of the ecosystem service providers (i.e., the species nodes directly connected to ecosystem service nodes) using the NetIndices R package v 1.4.4[58]. We used the iGraph R package v 1.2.5[59] to calculate the redundancy of each ecosystem service (Supplementary Table 1). Specifically, we define redundancy as and calculated it as the in-degree centrality—number of incoming links—for the ecosystem service nodes.

**Robustness analysis**. We develop a new approach that adapts and extends robustness analyses from food webs[15,17,50] to ecosystem services. We define ecosystem service robustness as the secondary loss of ecosystem services following primary species losses. To address our primary question, we compare the robustness of ecosystem services in aggregate (i.e., all seven ecosystem services) to that of the food web under a variety of species extinction sequences.

Following Bane et al.[29], food web robustness ($R_F$) is calculated as the Area Under the Curve (AUC, Supplementary Fig. 5), using $R_F = \frac{\Sigma y(x)}{max(x)}$, where $x$ is the proportion of target species directly removed (i.e., primary extinctions) and $y$ is the proportion of susceptible species that did not go secondarily extinct (Fig. 4, Supplementary Fig. 5a; see Supplementary Methods). Susceptible species include species that can go secondarily extinct in this bottom-up extinction model (i.e., nonbasal species). Therefore, basal species are not considered susceptible. Our species loss sequences did not all include the full species list, so we divided the AUC by the maximum proportion of species removed to scale robustness to the length of the sequence. We calculated food web robustness using the food web without ecosystem services.

Extending these analyses, we developed an approach to calculate ecosystem service robustness, using the food webs with ecosystem services. Specifically, we measured ecosystem service robustness by sequentially removing species and tracking the secondary loss of particular nodes—the nodes representing each of the seven ecosystem services. Ecosystem service robustness ($R_{ES}$) is calculated as the AUC (Supplementary Fig. 5b), using $R_{ES} = \frac{\Sigma y(x)}{max(x)}$, where $x$ is the proportion of target species nodes directly removed (i.e., primary extinctions) and $y$ is the proportion of ecosystem service nodes that were not secondarily lost (Fig. 4, Supplementary Fig. 6a, see Supplementary Methods). This calculation considers services to be lost when all ecosystem service providers are removed or lost, implying that all species contribute equally to a given ecosystem service—an assumption we relax below.

First, we considered ecosystem service nodes in aggregate, tracking the loss of all services as species are removed or lost secondarily ($R_{ES}$). Next, to address our secondary question, we modified the calculation of ecosystem service robustness ($R_{ES}$), to instead consider each of the seven ecosystem services individually. We measure and calculate individual ecosystem service robustness ($R_{indiv}$) in the same way as ecosystem service robustness ($R_{ES}$), but instead track when a single ecosystem service is secondarily lost (Supplementary Fig. 5c)—and calculate individual ecosystem service robustness as the AUC up until the ecosystem service is lost. Therefore, individual ecosystem service robustness ($R_{indiv}$) is calculated using $R_{indiv} = \sum x(y)$, where $x$ is the proportion of target species removed and $y$ is

a binary value that indicates whether a single ecosystem service remains in the network ($y = 1$) or not ($y = 0$, Supplementary Fig. 5c).

**Robustness extension: weighted contributions of species to ecosystem services**. Species often contribute to ecosystem services unequally, so we extend our analyses to consider species' varying contributions to each service. We extended the aggregate ($R_{ES}$) and individual ($R_{indiv}$) ecosystem service robustness methods to account for weighted contributions. In the original robustness calculation, the proportion of ecosystem services remaining ($y$) only decreased when an ecosystem service was completely lost (i.e., there were no ecosystem service providers remaining). Here, we used species' biomass data to track decreases in ecosystem services that result from the loss of individual ecosystem service providers[48,49] (see Supplementary Methods, Supplementary Fig. 6). Here, we assume that the relationship between ecosystem service providers' biomasses and ecosystem service amount is linear, and that the $y$-intercept of this relationship is the same for all ecosystem services and ecosystem service providers. We recalculated $R_{ES}$ and $R_{indiv}$ for all ecosystem services except birdwatching. We did not include birdwatching in this extension because the relationship between species and birdwatching is not proportional to species' biomass[46,55,60].

**Species loss sequences**. We simulated 12 sequences of species extinctions (Fig. 2). We supplemented extinction sequences from food web studies (e.g., most to least connected species[15,16,24,26,50]), with novel sequences of our own design, described next. These sequences varied in length (Supplementary Table 2).

To address Q3, we develop and apply novel sequences that remove species involved in ecosystem services: ecosystem service providers and supporting species (Fig. 1b). We then examine the robustness of both the food webs and ecosystem services to these extinction sequences. To test whether ecosystem service providers and their supporting species are also critical for food web robustness, we focus on the consequence for food web robustness from removing species directly critical for ecosystem services and those supporting ecosystem services indirectly.

Novel sequences of species loss: ecosystem service-centered sequences

1. Ecosystem service providers: We removed the species that directly provide each of the seven ecosystem services (see Section 2.2 and Supplementary Information). We ran three sequences where species lost were ordered from (1) high to low biomass, where species with high biomass were removed first, (2) low-to-high biomass, and (3) random losses, simulated 100 times (Supplementary Fig. 2, see Supplementary Methods). We also ran two sequences ordered by (1) high-to-low and (2) low-to-high abundance, which are not included in our final results (see Supplementary Methods).

2. Supporting species: To identify species that indirectly support ecosystem services, we first applied a personalized PageRank approach[61]. This method builds on Google's PageRank™ algorithm, which ranks web pages as "important" or relevant to user's searches[62]. PageRank models a random walker that pursues a path at random through a directed network (Supplementary Fig. 3b). Specifically, the random walker starts at a random node and walks a path, where the path terminates at a node with a fixed probability, $\alpha$, and continues to another random node with probability $1 - \alpha$ (known as the damping factor). When a path terminates, the process is repeated at a new, random starting node. For each node in a network, its PageRank score of importance is the probability that the walker visits that node[62]. Generally, if we consider a food web where species support other species through nutrient or energy transfers, a species is considered important if it facilitates (directly or indirectly) this transfer to other species[63].
Personalized PageRank allows us to specify the starting node in the random walk. This allows us to gauge a node's importance in relation to the starting node. Here, we start the random walks at ecosystem service nodes, so we can identify species that indirectly contribute to ecosystem services (i.e., which species are indirectly important to ecosystem services). To do this, we flipped the direction of our network, so that arrows pointed from ecosystem services to ecosystem service providers, and from consumers to prey (Supplementary Fig. 3a). Using the iGraph package v 1.2.5 in R[59], we applied the PageRank function ($\alpha = 0.15$, Supplementary Fig. 4) to model a personalized PageRank once for each service (Supplementary Fig. 3b). We ran seven personalized PageRank models, specifying a single service as the starting node each time, yielding seven different personalized PageRank values for each species in the food webs (see Supplementary Methods). Before calculating each species' mean indirect contribution (mean Personalized PageRank score) to all ecosystem services, we removed the species directly connected to each individual ecosystem service in our link list. This left us with a different list of species and personalized PageRank values for each service (i.e., a species that was removed for directly providing one service could remain in another list if it indirectly supports another ecosystem service). To rank species on their overall, indirect contributions to ecosystem services, we calculated the mean personalized PageRank score across all ecosystem services for each species. We ran three sequences of species loss for supporting species, two of which were based on the personalized PageRank score: (1) high-to-low, (2) low-to-high, and (3) random removal of supporting species (i.e., random with respect to the mean page rank scores).

Novel sequences of species loss: threat-based sequences

3. Rare species: Rare species are more likely to be lost from ecosystems in response to anthropogenic threats or demographic stochasticity[64,65] and can disproportionately contribute to functional diversity, ecosystem functioning, and resilience[28,66] (but see ref. [67]). For this extinction sequence, we sorted species using relative abundance as an indicator for rarity[64], which was calculated based on abundance (number of individuals/hectare) data from Hechinger et al.[52] Previous robustness studies have simulated species loss based on body size[20,42,68], which can influence extinction risk. Body size is correlated with abundance but is difficult to determine for plants. Instead, we removed species in ascending order of relative abundance (rarest first) to capture body-size effects of extinction risk while also including plants.

4. Species vulnerable to system-specific threats ("Vulnerable species"): We identified system-specific threats to salt marsh/estuary ecosystems and species vulnerable to these threats by conducting a literature synthesis in Google Scholar (see Supplementary Methods; Supplementary Data 2). We considered three stressors to estuaries: estuarine acidification (acid sulfate soils), eutrophication and hypoxia, and pollution. We systematically searched each species—threat combination and recorded a species as threatened if a reference existed to support that species displayed known mortality to the threat (see Supplementary Methods). Species with known mortality to any one of these three threats was included in this sequence. Our approach to designate a subset of species as vulnerable is conservative, given that these threats likely impact other species, but those impacts were not yet documented at the species level. This sequence is therefore much shorter than the others. The impact of a threat is likely a combination of population size and susceptibility to the threat, so we ordered this sequence by abundance, where less abundant species are removed first.

Commonly studied sequences: topological sequences

We next compared our novel threat-based and service-centered sequences to topological sequences used in other food web robustness studies (e.g., [15,20,24,25]). Specifically, we ran the following sequences of species loss.

5. Most-to-least connected sequence: We calculated species connectedness (i.e., node degree centrality, or the total number of links each species has) using the R package, iGraph v 1.2.5[59], and then removed species from most to least connected (i.e., most to least number of total links).

6. Random sequence: We simulated 1000 random extinction sequences following[12,16,25,39,44] and found the mean, highest (max.), and lowest (min.) robustness values. These values served as the average, as well as the best and worst-case scenarios, respectively.

All removal sequences were simulated in R (see Supplementary Methods).

**Correlation analysis**. To examine their relationship, we ran a Spearman's correlation test on the food web and ecosystem service robustness for all 3 systems and all 12 sequences described above. We ran three additional Spearman's correlation tests on robustness values grouped by the type of sequence: (1) topological sequences (most-to-least connected, random; $n = 12$), (2) threat-based sequences (rarity, vulnerable species; $n = 6$), and (3) ecosystem service sequences (ecosystem service providers, supporting species; $n = 18$).

**Regression analyses examining differences across services**. We fit seven linear regressions to examine whether variation in individual ecosystem service robustness is associated with the trophic level and redundancy (see Supplementary Methods). Specifically, we fit a model that included all extinction sequences, and a model for each of the following sequences: most-to-least connected, ecosystem service providers (high-to-low and low-to-high biomass), supporting species (most-to-least and least-to-most important), and rarity.

**Sensitivity tests**. We probed the sensitivity of our results to various modeling choices, including (1) the $x$ axis equation used to calculate robustness (based on the area under the curve, Supplementary Information Section I), (2) the number of ecosystem services considered (see Supplementary Discussion, Supplementary Tables 4–6, Supplementary Fig. 10), (3) the length of the species-removal sequences (see Supplementary Discussion, Supplementary Tables 2 and 3), (4) the inclusion of outliers in the correlation analyses, and, (5) additional interaction types, like parasitism, whose inclusion can alter robustness[41,69] (Supplementary Figs. 11 and 12, Supplementary Table 7; see Supplementary Methods and Discussion). We also ran three additional Spearman's correlation tests to test the relationship between (1) food web robustness ($R_{FW}$) and the weighted, aggregate ecosystem service robustness, (2) unweighted and weighted, aggregate ecosystem service robustness, and (3) unweighted and weighted, individual ecosystem service robustness.

**Reporting summary**. Further information on research design is available in the Nature Research Reporting Summary linked to this article.

## Data availability

Hechinger et al.[52] provide the published, unfiltered data on species interactions. We have included the filtered species and interaction data used in our analyses as Supplementary

Data Files 1 and 2. Xiao et al.[49] provide information on links between functional groups and ecosystem services. Species-level data connecting species and services, as used here, have been submitted as a data paper and are available upon request from the authors (A. K. and L.D.). We used the eBird database and Google Scholar to collect ecosystem service data. All data used to generate the results were available to reviewers.

## Code availability

The code used to generate the results of this paper is archived on GitHub (https://doi.org/10.5281/zenodo.4437248). Access to the source functions used in this code is available upon request from co-author, A.B. All codes, including source functions, were available to reviewers.

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

## Acknowledgements

We thank D. Larremore, J. Resasco, and K. Suding for comments on this research, and D. Larremore for input and guidance on PageRank methods. Publication of this article was funded by the University of Colorado Boulder Libraries Open Access Fund and the University of California Santa Barbara Open Access Publishing Fund. We also acknowledge support from NSF OCE #2049360 and #2049304 to L.D. and A.B.

## Author contributions

A.K. and L.D. designed the study with input from A.B. and J.M., A.K., and L.D. collected the ecosystem service data. A.B. provided the original R code for food web robustness, and A.B. and A.K. modified the code to extend the robustness analysis to ecosystem services. A.K. performed modeling work, sensitivity analyses, and analyzed the output data. A.K. and L.D. wrote the first draft of the paper, and all authors contributed to revisions.

## Competing interests

The authors declare no competing interests.
