## [Peer Review File · Nature Communications]

Reviewer comments, first round:

Reviewer #1 (Remarks to the Author):

This ms is very well written and certainly deserves publication in a high quality journal. It will spur further integration of food web and ES theory and practical application and lead to better understanding of interactions between the two.

Certainly, the writing and topic make this of general interest to economist or ecologists no matter the system they work in. The authors use three well studied food web systems to test their ideas and generality of those ideas across sites ES are also well worked out for marshes, so this is excellent system to look at interactions between the two.

My questions for the authors lie more in novelty and generality of the work and limitations.

1. From an empirical perspective there has been much work to look at how changes in food web structure affect ES. Estes has completed a lot of this work in kelp forests and there has been synthesis work completed on that. In their ecology papers, there are even food web models that link to changes in ES. So this exercise in general has been done before. This paper looks at robustness in both though and does more general tests. Does the take away from the novelty?

2. Southern California Marshes are one of the few marshes in the world where the foundation species are not eaten by marsh grazers. In fact, most marine systems in the world have ample grazers and top down forcing in the system. So rather than being a representative system, this one is likely to over estimate the importance of bottom up linkages and underestimate the importance of top down forcing in driving changes in food webs and ES. This is mentioned in a small way in discussion, but much greater discussion is needed. In fact, it would be great if food web data from a marsh with top down control is added and these ideas tested there. I think Jarrett Byrnes is collecting data on that, and he might be good to contact.

3. Non trophic interactions are incredible powerful in salt marshes and dominate the generation of ES, in fact so do parasites. Both of these forces are not included in this iteration network and to get true understanding of impacts of species losses, one really does need to include these in some way or form. Or talk about how not including them could drastically underestimate how food web changes impact ES changes.

Reviewer #2 (Remarks to the Author):

The study by Keyes and co-workers addresses how secondary extinction cascades can indirectly affect ecosystem services. They use a topological approach to predict extinctions in 12 sequences applied to three intertidal food webs. This is a very interesting research topic, and I found the introduction very well written to lay out the guiding questions. Similarly, the rest of the manuscript has a clear structure, is easy to follow, and a pleasure to read (and review). I have one major issue with the methodology that is used and some minor points that are also listed below. Although I know that critical points are not always perceived as helpful, it is my attention to help this publication increasing its breadth and impact. Overall, I very much enjoyed reading this manuscript.

A topological approach predicts secondary extinctions. This approach implies that species can go secondarily extinct only if they lose all of their resources. Indeed, this method is still wide-spread in food-web ecology, but it has several severe limitations. First, top-down effects are specifically excluded. A trophic cascade arising from a top predator's primary extinction would not occur in this type of model. Second, lateral effects cascading through the network are also excluded. Third, many of the ecosystem services described in the paper are – in my opinion – density-dependent,

and non-lethal changes in density are not covered by this method. If the loss of a critical resource reduced the biomass density of a target species by 95%, the associated service would decrease substantially, but this would not occur in the model if an alternative resource is present. Fourth, the secondary extinction sequences are very different depending on whether you use a topological approach (as in this study) or an approach with population dynamics that included bottom-up and top-down effects (see Curtsdotter et al. 2011, cited in the manuscript). The current discussion (lines 436-445) on dynamical approaches is focused on adaptive re-wiring and does not address these points. I agree that there is currently limited information on how populations would re-wire their linkage pattern following a primary species loss but a dynamical approach without re-wiring would be possible. As population dynamic packages are available in R, I would suggest switching to a dynamic approach. If this is not possible, I would at least expect a discussion of the limitations of the approach chosen and why the alternative has not been employed.

Minor points:

Lines 137-142: The original food webs were modified by excluding parasites and aggregating life stages. It would help the reviewers and readers if the modified food-web matrices, including the trait information, would be published in the supplement. As soon as you start thinking about replicating this model approach, the details become tricky. For instance, I would not be sure what body size or abundance is associated with aggregated life stages.

Lines 211-219: I could not follow the procedure of how the robustness of ecosystem services is calculated. Based on the flow of arguments that it follows the same idea as the calculation of food-web robustness, I expect that everything should be ok, but this method's description needs more detail. Does the description imply that are associated with an ecosystem service contribute equally? Or is it weighed by abundances? As a first guess, I would argue that the latter makes more sense, but a weighting by biomass density might be ideal. For instance, the services of carbon sequestration, fishery, shoreline stabilization, water filtration, and wave attenuation should all scale relative to the biomass density of the population providing the service. A more detailed description is mandatory before it can be discussed if the algorithm is reasonable.

[Redacted]

We have decided to retain the topological approach (see rationale below in responses to Reviewer 1's comments and email correspondence with Reviewer 2), but have expanded it to include species' weighted contributions to ecosystem services (see details below and highlighted sections in the revised manuscript).

Reviewer(s)' comments to Authors:

Reviewer: 1

This ms is very well written and certainly deserves publication in a high-quality journal. It will spur further integration of food web and ES theory and practical application and lead to better understanding of interactions between the two. Certainly, the writing and topic make this of general interest to economist or ecologists no matter the system they work in. The authors use three well studied food web systems to test their ideas and generality of those ideas across sites ES are also well worked out for marshes, so this is excellent system to look at interactions between the two.

We thank the reviewer for their kind comments about our manuscript and its contribution.

My questions for the authors lie more in novelty and generality of the work and limitations.

1. From an empirical perspective there has been much work to look at how changes in food web structure affect ES. Estes has completed a lot of this work in kelp forests and there has been synthesis work completed on that. In their ecology papers, there are even food web models that link to changes in ES. So this exercise in general has been done before. This paper looks at robustness in both though and does more general tests. Does this take away from the novelty?

Our paper primarily aims to quantify whether ecosystem services are more or less robust to species losses than the food webs that support them – a key, unanswered question in the ecology of ecosystem services. While Estes et al. similarly study the impact of cascading extinctions on ecosystem services, they use smaller species interaction networks (e.g., 7 species¹), and generally only consider top-down effects, rather than losses in resources for consumers. Our approach incorporates a higher degree of complexity and is less focused on the link between structure and function. Instead, our work focuses on food web and ecosystem service stability; which has commonly been analyzed using robustness analyses for almost 20 years²⁻⁴. However, to better situate the work in the field, we added more details on top-down dynamics (L240-243) and added an Estes et al. citation on L242: “Estes, J. A. et al. Trophic downgrading of planet earth. Science (2011). doi:10.1126/science.1205106”

2. Southern California Marshes are one of the few marshes in the world where the

foundation species are not eaten by marsh grazers. In fact, most marine systems in the world have ample grazers and top down forcing in the system. So rather than being a representative system, this one is likely to overestimate the importance of bottom up linkages and underestimate the importance of top down forcing in driving changes in food webs and ES. This is mentioned in a small way in discussion, but much greater discussion is needed. In fact, it would be great if food web data from a marsh with top down control is added and these ideas tested there. I think Jarrett Byrnes is collecting data on that, and he might be good to contact.

As Dr. Laura Dee (coauthor) is a collaborator of Dr. Jarrett Byrnes, we have reached out to Dr. Byrnes and confirmed the data is not yet finished, but that this is a fruitful direction for future work and collaboration. As Byrnes' data is not yet published or publicly available, we cannot perform analyses on the systems that the reviewer mentioned. However, in response, we expanded our discussion to include additional details about top-down dynamics (L240-243) and a follow-up paragraph that outlines our reasoning for using a topological approach (L260-288).

To elaborate, a dynamic food web modeling approach would be needed to capture these top-down effects^{5,6}. However, we opted to use a topological approach, rather than a dynamic modeling approach, for several reasons. The primary one is that there are many challenges and uncertainties associated with parameterizing dynamic models of food webs (in terms of parameters, functional forms, etc.) given existing data, which can dictate the dynamics⁶. Further, robustness studies that do incorporate dynamics typically use simulated data, as opposed to empirical data^{5,7-9}. One benefit of a topological approach, as we have done here, is that it doesn't require those strong assumptions about many largely unknown parameters^{6,10}. This allows us to model secondary extinctions on more complex, larger (more species), and real systems that include services. Demonstrating these methods, concepts and questions for ecosystem services in *real systems* was one of the contributions of this paper, which we clarify in the paper (L268-270) – and key part of its contribution as you point out in your review above.

Another key reason for starting off with the topological approach is the complexity of dynamics between species and services. We see incorporating ecosystem service dynamics as an important frontier in ecosystem service science (and food web modeling; L270-272); and as such, we believe that topic merits its own paper(s). In particular, very few ecosystem service studies that consider the underlying ecology do not incorporate dynamics, particularly in a multi-species setting. Generally, species' or functional groups' contributions to ecosystem services are not known, except for services that depend on population harvest (L274-276). For example, we know species contributions for fisheries, where the service is measured and valued based on biomass of particular species, like cod. On the other hand, for services like water filtration or shoreline protection, species-specific, direct contributions and dynamics of those relationships through time are usually unclear, and can be highly complex (e.g., dependent on diversity, abundance, and identities of species). Outside of the fisheries realm, ecosystem service assessments tend to measure these types of services by lumping species together based on habitat type using land cover data (reviewed in^{11,12}). For these reasons, we opted to use a topological approach, which doesn't require strong assumptions about these unknown dynamics for ecosystem services. That being said, the prior version of the paper only weighted species equally in their contribution (e.g., akin to studies that examine 'species richness' and its relationship with

services). However, previous studies have also weighted species' contributions based on their biomass or abundance^{13,14}. Following the suggestion of R2, we have expanded our approach to consider and measure robustness of ecosystem services when species contribute unequally based on their biomass (see below; L167-174, L278-286, L406-416, L536-539).

See references at the end of the document.

3. Non trophic interactions are incredible powerful in salt marshes and dominate the generation of ES, in fact so do parasites. Both of these forces are not included in this iteration network and to get true understanding of impacts of species losses, one really does need to include these in some way or form. Or talk about how not including them could drastically underestimate how food web changes impact ES changes.

Robustness analyses, and our extensions to ecosystem services, can be applied to any form of interactions, or multiple. We have clarified the generality of this approach for incorporating multiple interaction types in the discussion (L185-187): “The methods used in this study are applicable to a variety of trophic interactions (e.g., predation, micropredation, parasitism) and ecological networks.”

To demonstrate, as we agree with the review about the importance of parasites in salt marshes, we now performed the analyses with and without the inclusion of parasites and their parasitic interactions. We chose to include parasitism as another form of interaction, rather than other forms of interactions, since they are well-resolved in these systems, whereas the other interaction types are not available in the existing dataset. The addition of non-trophic interactions to food webs is the leading edge of network ecology, with very few existing datasets that quantitatively incorporate multiple interaction types¹⁵. We have now added revisions to the methods (L534-536), discussion, and Supplementary Methods-Results, including newly added figures and tables (see Supplementary Figures 11-12 and Table 8) in response. We next summarize these revisions.

To test the sensitivity of our results to the exclusion of parasites and parasitic interactions, we re-ran our robustness analyses for the food webs and ecosystem services for all sequences on networks that included parasites and parasitic interactions. By including parasites and their interactions, the food webs increased in size (in both number of species and links) across all three salt marshes¹⁶. Table S8 outlines these metrics for each of the webs with parasites, services, and parasitic interactions.

Our main results were not sensitive to including parasitism. When we included parasites and their interactions, food web robustness remained strongly and positively correlated with ecosystem service robustness ($r_s[36]=0.738$, $p=2.8e-07$; see Figure S11). There was slight variation in food web and ecosystem service robustness values, where they tended to be higher when parasites are included (Supplementary Figure 12). This is likely due to the fact that we included adult life stages only, which can inflate robustness^{17,18}. However, similar to previous studies that have compared food web robustness with and without parasites on these food webs, robustness decreased for both food webs and ecosystem services when parasites were included for two sequences: most to least connected¹⁹ and random^{18,19}.

Reviewer: 2

The study by Keyes and co-workers addresses how secondary extinction cascades can indirectly affect ecosystem services. They use a topological approach to predict extinctions in 12 sequences applied to three intertidal food webs. This is a very interesting research topic, and I found the introduction very well written to lay out the guiding questions. Similarly, the rest of the manuscript has a clear structure, is easy to follow, and a pleasure to read (and review). I have one major issue with the methodology that is used and some minor points that are also listed below. Although I know that critical points are not always perceived as helpful, it is my attention to help this publication increasing its breadth and impact. Overall, I very much enjoyed reading this manuscript.

We thank the reviewer for their positive and thoughtful comments and appreciate their willingness to provide additional resources and feedback on our manuscript. As the reviewer offered to discuss the manuscript and reviews with us, we have included this fruitful exchange in a separate document.

A topological approach predicts secondary extinctions. This approach implies that species can go secondarily extinct only if they lose all of their resources. Indeed, this method is still wide-spread in food-web ecology, but it has several severe limitations. First, top-down effects are specifically excluded. A trophic cascade arising from a top predator's primary extinction would not occur in this type of model. Second, lateral effects cascading through the network are also excluded. Third, many of the ecosystem services described in the paper are – in my opinion – density-dependent, and non-lethal changes in density are not covered by this method. If the loss of a critical resource reduced the biomass density of a target species by 95%, the associated service would decrease substantially, but this would not occur in the model if an alternative resource is present. Fourth, the secondary extinction sequences are very different depending on whether you use a topological approach (as in this study) or an approach with population dynamics that included bottom-up and top-down effects (see Curtsdotter et al. 2011, cited in the manuscript). The current discussion (lines 436-445) on dynamical approaches is focused on adaptive re-wiring and does not address these points. I agree that there is currently limited information on how populations would re-wire their linkage pattern following a primary species loss but a dynamical approach without re-wiring would be possible. As population dynamic packages are available in R, I would suggest switching to a dynamic approach. If this is not possible, I would at least expect a discussion of the limitations of the approach chosen and why the alternative has not been employed.

We agree that using a topological approach has several limitations compared to dynamic approaches, and with the reviewer that we should further elaborate on those limitations to present a balanced view of the field. In response, we did two main revisions. First, we clarify why we used this approach in greater detailed, including the lack of well-resolved relationships on the dynamics between particular species and most services (L260-288). We discussed our rationale for why opted to use this approach for a few reasons with the reviewer 2 and have included this very fruitful email correspondence in a separate document.

Second, after corresponding with the editor and R2, and deciding to retain the topological approach, we have added additional discussion of the importance of top-down dynamics and lateral effects (points 1 and 2) to our Discussion paragraph on the limitations (L234-237, L240-243, L250-253); and further emphasize the previously written text about top-down dynamics (L238-240). We also added a Discussion paragraph that explains why we chose to use the topological approach, as opposed to the dynamic approach the reviewer mentions (L260-288), including elaborating on the knowledge gaps surrounding links between species, ecosystem services from food webs, and dynamics.

Related, to address the reviewer's third point, we added a method (L406-416, L536-539) and results (L167-174) extension to test how ecosystem services robustness varies when ecosystem service provisioning is *weighted* based on species' biomass compared to our unweighted calculation. Taking this weighted approach, we found that our main results (i.e., food web and ecosystem service robustness are correlated) did not change, but that ecosystem service robustness did decrease when weights were incorporated. This is because each time an ecosystem service provider is lost, the proportion of ecosystem services remaining decreases. In the unweighted calculation, the proportion of ecosystem services remaining only decreases when it loses all of its ecosystem service providers. We augmented Figure 6A to show the variation in individual ecosystem service robustness with and without weights and included two new figures (Supplementary Figures 11-12). For more details, also see the minor point #2 below.

Minor points:

Lines 137-142: The original food webs were modified by excluding parasites and aggregating life stages. It would help the reviewers and readers if the modified food-web matrices, including the trait information, would be published in the supplement. As soon as you start thinking about replicating this model approach, the details become tricky. For instance, I would not be sure what body size or abundance is associated with aggregated life stages.

We fully agree. We are working on submitting a data paper with all data, meta-data, and procedures to *Ecology*, to publish this data. As the paper is in review, we have included these files herein (in the .zip folder) for the reviewer and editor to review. If the manuscript is accepted, we can work out how to publish the data if the Ecology data paper is still pending, to ensure it is available.

To address the comment about aggregated life stages, we revised the text on line 142 to more accurately state the methods used: "...filtered the species to include only adult life stages..." (L313-315). The consequences of this step are addressed in the Discussion (L253-256) and the Supplementary Results.

Lines 211-219: I could not follow the procedure of how the robustness of ecosystem services is calculated. Based on the flow of arguments that it follows the same idea as the calculation of food-web robustness, I expect that everything should be ok, but this method's description needs more detail. Does the description imply that are associated with an ecosystem service contribute equally? Or is it weighed by abundances? As a first guess, I would argue that the latter makes more sense, but a weighting by biomass density might be

ideal. For instance, the services of carbon sequestration, fishery, shoreline stabilization, water filtration, and wave attenuation should all scale relative to the biomass density of the population providing the service. A more detailed description is mandatory before it can be discussed if the algorithm is reasonable.

Thank you for pointing out the lack of clarity. We have addressed this comment in two ways. First, we added a sentence to the specified paragraph on the ecosystem service robustness calculation to clarify how it is calculated (L391-393): “This calculation considers services to be lost when all ecosystem service providers are removed or lost, implying that all species contribute equally to a given ecosystem service – an assumption we relax below.” Second, we added a method (copied below; L406-416, L536-539) and results (L167-174) extension to test how ecosystem services robustness varies when ecosystem service provisioning is weighted based on species’ biomass compared to our unweighted calculation.

“Robustness extension: Weighted contributions of species to ecosystem services

Species often contribute to ecosystem services unequally, so we extend our analyses to consider species’ varying contributions to each service. We extended the aggregate (R_{ES}) and individual (R_{indiv}) ecosystem service robustness methods to account for weighted contributions. In the original robustness calculation, the proportion of ecosystem services remaining (y) only decreased when an ecosystem service was completely lost (i.e., there were no ecosystem service providers remaining). Here, we used species’ biomass data to track decreases in ecosystem services that result from the loss individual ecosystem service providers^{31,43} (see SI, Figure S6). We re-calculated R_{ES} and R_{indiv} for all ecosystem services except birdwatching. We did not include birdwatching in this extension because the relationship between species and birdwatching is not proportional to species’ biomass^{34,44,45}.”

References

1. Estes, J. A., Steneck, R. S. & Lindberg, D. R. Exploring the consequences of species interactions through the assembly and disassembly of food webs: A pacific-atlantic comparison. *Bull. Mar. Sci.* **89**, 11–29 (2013).
2. Dunne, J., Williams Richard J & Martinez, N. D. Network structure and biodiversity loss in food webs: robustness increases with connectance. *Ecol. Lett.* **5**, 558–567 (2002).
3. Dunne, J. A., Williams, R. J. & Martinez, N. D. Network structure and robustness of marine food webs. *Mar. Ecol. Prog. Ser.* **273**, 291–302 (2004).
4. Montoya, D., Yallop, M. L. & Memmott, J. Functional group diversity increases with modularity in complex food webs. *Nat. Commun.* **6**, (2015).
5. Curtsdotter, A. *et al.* Robustness to secondary extinctions: Comparing trait-based sequential deletions in static and dynamic food webs. *Basic Appl. Ecol.* **12**, 571–580 (2011).
6. Eklöf, A., Tang, S. & Allesina, S. Secondary extinctions in food webs: A Bayesian network approach. *Methods Ecol. Evol.* **4**, 760–770 (2013).
7. Gross, K. & Cardinale, B. J. The functional consequences of random vs. ordered species extinctions. *Ecol. Lett.* **8**, 409–418 (2005).
8. Eklöf, A. & Ebenman, B. Species loss and secondary extinctions in simple and complex model communities. *J. Anim. Ecol.* **75**, 239–246 (2006).
9. Dunne, J. A. & Williams, R. J. Cascading extinctions and community collapse in model food webs. *Philos. Trans. R. Soc. B Biol. Sci.* **364**, 1711–1723 (2009).
10. De Visser, S. N., Freymann, B. P. & Olf, H. The Serengeti food web: Empirical quantification and analysis of topological changes under increasing human impact. *J. Anim. Ecol.* **80**, 484–494 (2011).
11. Seppelt, R., Dormann, C. F., Eppink, F. V., Lautenbach, S. & Schmidt, S. A quantitative review of ecosystem service studies: Approaches, shortcomings and the road ahead. *J. Appl. Ecol.* (2011). doi:10.1111/j.1365-2664.2010.01952.x
12. Martnez-Harms, M. J. & Balvanera, P. Methods for mapping ecosystem service supply: A review. *International Journal of Biodiversity Science, Ecosystem Services and Management* (2012). doi:10.1080/21513732.2012.663792
13. Balvanera, P., Kremen, C. & Martínez-Ramos, M. Applying community structure analysis to ecosystem function: Examples from pollination and carbon storage. *Ecol. Appl.* (2005). doi:10.1890/03-5192
14. Xiao, H. *et al.* Win-wins for biodiversity and ecosystem service conservation depend on the trophic levels of the species providing services. *J. Appl. Ecol.* **55**, 2160–2170 (2018).
15. Hutchinson, M. C. *et al.* Seeing the forest for the trees: Putting multilayer networks to work for community ecology. *Functional Ecology* (2019). doi:10.1111/1365-2435.13237
16. Dunne, J. A. *et al.* Parasites Affect Food Web Structure Primarily through Increased Diversity and Complexity. *PLoS Biol.* **11**, 1–17 (2013).
17. Lafferty, K. D. *et al.* Parasites in food webs: the ultimate missing links. *Ecol. Lett.* **11**, 533–46 (2008).
18. Rudolf, V. H. W. & Lafferty, K. D. Stage structure alters how complexity affects stability of ecological networks. *Ecol. Lett.* **14**, 75–79 (2011).
19. Lafferty, K. D. & Kuris, A. M. Parasites reduce food web robustness because they are sensitive to secondary extinction as illustrated by an invasive estuarine snail. *Philos. Trans. R. Soc. B Biol. Sci.* **364**, 1659–1663 (2009).

Reviewer comments, second round:

Reviewer #1 (Remarks to the Author):

The authors have done a very nice job addressing concerns of reviewers and the ms is still very nice and of general interest. They still need to address one more major issue in relation to the generality of their major conclusion that states:

"Further, we find that ecosystem service providers do not play a critical role in stabilizing food webs – whereas species playing supporting roles in ecosystem services are critical to the robustness of both food webs and ecosystem services. "

Here is the question, since southern California is an anomaly among marsh areas globally in not having strong topdown control, not having strong secondary foundation species effects on eco. function, and not having strong ecosystem engineering effect, how general will this big conclusion be - even for marshes?

Angelini, C., et al 2016. A keystone mutualism underpins resilience of a coastal ecosystem to drought. *Nature Communications* 7:12473.

Angelini, C. et al. 2015. Foundation species' overlap enhances biodiversity and multifunctionality from the patch to landscape scale in southeastern United States salt marshes. *Proceedings of the Royal Society of London B* 282(1811): 2015.0421.

He, Q., and B. R. Silliman. 2016. Consumer control as a common driver of coastal vegetation worldwide. *Ecological Monographs* 86:278-294.

Bertness, M. Fiddler crab regulation of *Spartina* Growth. 1984, *Ecology*.

Bertness, M. Ribbed mussels regulation of *Spartina* Growth. 1986, *Ecology*.

Hensel, M. J, and B. R. Silliman. 2013. Consumer diversity across kingdoms supports multiple functions in a coastal ecosystem. *Proceedings of the National Academy of Sciences* 110:20621-20626.

Reviewer #2 (Remarks to the Author):

I have been reading this revised manuscript with great pleasure. I think it is one of the indicators of a great manuscript that any time I am reading it, there are new ideas emerging. I am sure that this will be a very influential contribution to the field. As we have been engaged in email contact, my comments have been addressed in a very satisfactory way. I have very much enjoyed this interaction and hope that we can continue review processes in this way. I have a couple of minor points that are listed below, but I am sure that they are by no means obstacles on the path towards publication.

Minor points

- 1) The supplement is missing the page numbers.
- 2) I very much like the addition of the analysis of weighted species contributions. I suggest to add that this implies (i) linear relationships between biomass and ES, and (ii) the same intercept of the relationship between ES and biomass across all species and ES. There is nothing wrong with these assumptions, they should just be made explicit.
- 3) It would be great to have a bit more explanation how the biomasses were estimated for species without biomass data in the original publication.

Signed, Ulrich Brose

We thank both reviewers for their comments and suggestions, which have improved the clarity of our manuscript and our study's implications.

Below, reviewer comments are in **bold**, and author responses are in normal font.

Reviewer(s)' comments to Authors:

Reviewer: 1

The authors have done a very nice job addressing concerns of reviewers and the ms is still very nice and of general interest. They still need to address one more major issue in relation to the generality of their major conclusion that states:

"Further, we find that ecosystem service providers do not play a critical role in stabilizing food webs – whereas species playing supporting roles in ecosystem services are critical to the robustness of both food webs and [SEP] ecosystem services. "

Here is the question, since southern California is an anomaly among marsh areas globally in not have strong topdown control, not having strong secondary foundation species effects on eco. functon, and not having strong ecosystem engineering effect, how general will this big conclusion be - even for marshes?

Angelini, C., et al 2016. A keystone mutualism underpins resilience of a coastal ecosystem to drought. Nature Communications 7:12473.

Angelini, C. et al. 2015. Foundation species' overlap enhances biodiversity and multifunctionality from the patch to landscape scale in southeastern United States salt marshes. Proceedings of the Royal Society of London B 282(1811): 2015.0421.

He, Q., and B. R. Silliman. 2016. Consumer control as a common driver of coastal vegetation worldwide. Ecological Monographs 86:278-294.

Bertness, M. Fiddler crab regulation of Spartina Growth. 1984, Ecology.

Bertness, M. Ribbed mussels regulation of Spartina Growth. 1986, Ecology.

Hensel, M. J, and B. R. Silliman. 2013. Consumer diversity across kingdoms supports multiple functions in a coastal ecosystem. Proceedings of the National Academy of Sciences 110:20621-20626.

We thank you for your comment, and for providing additional references. We have now explicitly addressed this limitation in our Discussion (L243-247), citing He et al. (2016): "The omission of top-down control could also have implications for the importance of ecosystem service providers in stabilizing food webs. Some salt marshes – with the exception of those in our study – are strongly controlled by top-down regulation³⁸, which could imply that higher

trophic level ecosystem service providers are more important in stabilizing food webs than we find here.”

Reviewer: 2

I have been reading this revised manuscript with great pleasure. I think it is one of the indicators of a great manuscript that any time I am reading it, there are new ideas emerging. I am sure that this will be a very influential contribution to the field. As we have been engaged in email contact, my comments have been addressed in a very satisfactory way. I have very much enjoyed this interaction and hope that we can continue review processes in this way. I have a couple of minor points that are listed below, but I am sure that they are by no means obstacles on the path towards publication.

We thank Reviewer 2, Dr. Brose, for his thoughtful and encouraging feedback throughout the revision process.

Minor points

1) The supplement is missing the page numbers.

Thank you for catching this, we have added page numbers to the supplement.

2) I very much like the addition of the analysis of weighted species contributions. I suggest to add that this implies (i) linear relationships between biomass and ES, and (ii) the same intercept of the relationship between ES and biomass across all species and ES. There is nothing wrong with these assumptions, they should just be made explicit.

This is a great point to highlight. We have added a sentence stating these assumptions in our Methods (L418-420): “Here, we assume that the relationship between ecosystem service providers’ biomasses and ecosystem service amount is linear; and that the y-intercept of this relationship is the same for all ecosystem services and ecosystem service providers.”

3) It would be great to have a bit more explanation how the biomasses were estimated for species without biomass data in the original publication.

We have added detail on how we estimated species biomasses in the Supplemental Information, Section I.4: “In the original dataset¹ not all species received a biomass estimate. We wanted to estimate the missing values for our analyses. In the original dataset biomass was estimated as the product of abundance and body size. All consumer species in the original data set had body size estimates, but only a subset had abundance estimates. We used published estimates on the correlation between abundance and body size in each estuary⁸ to estimate the missing abundance values. We then calculated the missing biomass values as the product of abundance and body size.”